# *In vitro* and *in silico* pharmaco-nutritional assessments of some lesser-known Nigerian nuts: *Persea americana*, *Tetracarpidium conophorum*, and *Terminalia catappa*

Efah Denis Eyong[1], Iwara Aripko Iwara[1], Eyuwa Ignatius Agwupuye[1], Abdulhakeem Rotimi Agboola[1], Daniel Ejim Uti[2,3]* Wilson Arong Obio[1] Esther Ugo Alum[2], Item Justin Atangwho[1]*

1 Department of Biochemistry, Faculty of Basic Medical Science, College of Medical Sciences, University of Calabar, P.M.B, Calabar, Nigeria, 2 Department of Research and Publications, Kampala International University, Kampala, Uganda, 3 Department of Biochemistry, Faculty of Basic Medical Sciences, Federal University of Health Sciences, Otukpo, Benue, Nigeria.

* dratangwho@gmail.com (IJA); daniel.ejimuti@kiu.ac.ug (DEU)

## Abstract

Together with their nutritional qualities, the biosafety, antidiabetic, antioxidant, and anti-inflammatory effects of *Tetracarpidium conophorum* nuts, *Persea americana* seeds, and *Terminalia cattapa* kernels were evaluated *in vitro* and *in silico.* RBC membrane stabilisation for anti-inflammatory characteristics, antioxidant activities by ABTS, DPPH, H2O2, and nitric oxide scavenging assays, and α-glucosidase and α-amylase inhibitory assays conducted in vitro were used to evaluate the anti-diabetic activity. With an IC50 value of 208 μg/mL, *P. americana* showed the maximum amount of inhibition, according to the results, while T. cat-appa showed a somewhat lower degree of inhibition at 236 μg/mL. *P. americana* exhibited the highest degree of α-amylase inhibition, with an IC50 value of 312 μg/mL. *T. catappa* showed the strongest DPPH radical scavenging activity, while T. conophorum showed the highest ABTS radical scavenging activity. *T. catappa* showed the strongest effectiveness in neutralising hydrogen peroxide. In tests using human red blood cells, *T. catappa* showed the strongest inhibition of RBC hemolysis. While *P. americana* showed higher concentrations of copper, manganese, potassium, and calcium, *T. catappa* showed higher magnesium concentrations. *T. catappa* had considerably higher levels of ash, proteins, lipids, and carbohydrates than *T. conophorum*, which had the highest quantity of crude fibre, according to proximate analysis. Molecular docking experiments have revealed that plant extracts from *P. americana,T. conophorum*, and *T. catappa* have substantial binding affinities towards α-glucosidase and amylase. Pseudococaine, M-(1-methylbutyl) phenylmethylcarbamate, o-xylene, and 1-deoxynojirimycin were the four compounds that showed binding affinities that were higher than those of acarbose. Acarbose and nitrate were not as compatible with docking scores as compared to the compounds dimethyl phthalate, pseudococaine, M-(1-Methylbutyl)phenyl methylcarbamate, 2-chloro-3-oxohexanedioic acid, and methyl 2-chloro-5-nitrobenzoate. These results suggest that these plant extracts hold great potential for the creation of thera-peutic medications that specifically target oxidative stress-related diseases like diabetes.

**Data availability statement:** Data is attached to this submission as supplementary files

**Funding:** The author(s) received no specific funding for this work.

**Competing interests:** We have no competing interests to disclose.

**Abbreviations:** ABTS, 2,2'-Azino-bis(3-ethylbenzothiazoline-6-sulfonic acid) radical, GC-MS, Gas Chromatography-Mass Spectrometry, DPPH, 2,2-diphenyl-1-picrylhydrazyl; $H_2O_2$, Hydrogen peroxide, RBC, Red blood cell; AOAC, Association of Official Analytical Chemists.

## 1. Introduction

Diabetes mellitus is a persistent metabolic and endocrine disease resulting in an elevated level of glucose in the blood, attributed to either reduced levels of insulin or resistance to insulin [1]–[3]. The therapeutic care of diabetes mellitus has faced significant difficulties in effectively controlling hyperglycemia, especially after mealsAmylase and α-glucosidase, which catalyse the hydrolysis of carbohydrates in the gastrointestinal tract, regulate postprandial hyperglycemia [4, 5]. According to Vadivelan et al. [6], reducing α-glucosidase and α-amylase expression slows glucose digestion and absorption. This reduces postprandial hyperglycemia in those with and without high blood sugar. Ijarotimi, [7] classed diabetes as inflammatory. Antioxidants and anti-inflammatories are vital to diabetic control. International Diabetes Federation classifies diabetes as T1DM and T2DM clinically. The cellular damage in T1DM sometimes causes complete insulin insufficiency. Asians and Africans use natural plant-based therapies for diabetes and other medical disorders due to the greater cost, limited availability, and negative effects of contemporary drugs [8]–[10].

Traditional herbal therapy has consistently remained a widely favoured method of healthcare. While herbal and conventional pharmacological therapies may have some variations, it is possible to assess the effectiveness and safety of herbal medications using traditional experimental techniques. Several distinct herbal extracts have demonstrated efficacy in treating particular medical ailments [11]–[13]. The seeds of *Persea americana (P. americana)* have been found to have notable hyperglycemic properties, which successfully reduce blood sugar levels and thus exhibit antidiabetic benefits. The observed impact is ascribed to many elements, including magnesium, calcium, potassium, sodium, and others, which have a pivotal function in maintaining blood homeostasis [14].

*T. conophorum* is a kind of seed that is cultivated primarily for its delectable taste and is suitable for consumption. The nut is mostly distributed in the forested areas of Africa and India [15]. *T. conophorum* is a kind of seed that is cultivated primarily for its delectable taste and is safe for consumption. They are members of the Euphorbiaceae family [16]. Nuts are eaten as snacks and drinks during the season [15]. The seeds are utilised for the treatment of infertility [17] and possess antimicrobial characteristics [17]. *Terminalia catappa* (*T. catappa*) can decrease the increase in insulin and glucose levels in the body following meals. It provides defence against detrimental organisms in blood glucose, a condition frequently experienced by individuals with diabetes when consuming foods with suddenly elevated sugar content [18, 19]. In addition to evaluating the extracts from these plants' probable modes of action by molecular docking analysis, The present research design intends to evaluate the *in vitro* effects of the extracts on diabetes, oxidative stress, and inflammation.

## 2. Method

### 2.1. Collection and identification of plants

Healthy seeds of Avocado (Persea americana) and African walnut (Tetracarpidium conophorum) bought at Watt Market in Calabar South, Cross River State, Nigeria. Additionally, *Terminalia catappa* was obtained from the University of Cross River State. The identification and certification of the seeds were carried out at the Department of Plant and Ecological Studies, University of Calabar. Subsequently, the plant materials were washed, followed by 7 days of air-drying. Afterwards, they were finely pulverised using an electric blender (Binatone - 56). Each batch of 500g of plant material was extracted with 200 ml of 95% ethanol for 24 hours using the cold percolation method. The ethanol extract was initially filtered through cheesecloth, then using filter paper. The filtrate was then concentrated using a rotary evaporator

under low pressure, and the ethanol was evaporated with a water bath. The resulting fresh paste was stored in an airtight container and refrigerated until needed.

## 2.2. Estimation of α-glucosidase inhibitory activity

To measure α-glucosidase inhibitory activity, we used a slight modification of the standardised procedure from Mopuri et al. (2008). In a nutshell, we filled 96-well plates with 50 microliters of phosphate buffer (pH 6.8) containing an α-glucosidase solution (0.2 microliters per millilitre) and 60 microliters of the sample. The mixture was incubated at 37 degrees Celsius for twenty minutes. After pre-incubation for some time, a 50μl solution of P-nitrophenyl-α-D-glucopyranoside (PNPG) in 0.1M phosphate buffer (pH 6.8) was added to the wells. The combined mixtures was subjected to incubation at a temperature of 37°C for an extra duration of 20 minutes. To stop the reaction, a 0.2M NaCO3 solution of 160μl was added to each well. The absorbance of the sample was taken at 405nm and compared to that of a standard sample.

## 2.3. Estimation of α-amylase inhibitory activity

The protocol outlined by Nasir et al. [20] was adopted to evaluate the α-amylase inhibitory characteristics. A total of 260 μl was used for the test, consisting of 40 μl of PBS (0.02M, pH 6.9), 100 μl of GOD reagent, 40 μl of soluble starch (2 g/L), the inhibitor solution, and the enzyme solution (2 units/mL). In summary, the plant extract and enzyme solution were combined and pre-incubated for 10 minutes at 37°C. After adding the starch solution and continuing to incubate for fifteen more minutes, the reaction beganFollowing an additional 15-minute incubation period, 100 μL of the GOD reagent was introduced. Subsequently, the absorbance was quantified at a wavelength of 505 nm. Measurements of absorbance were made with the samples and PBS acting as a negative control. A blank sample was provided for each concentration of the samples. For the kinetic studies, the GOD reagent was promptly added following the addition of starch, and measurements were recorded every minute for 45 minutes at a wavelength of 505nm. The results were measured using the IC50 value.

## 2.4. Estimation of ABTS radical scavenging activity

We evaluated the plant extract's radical-eliminating ability using the ABTS technique, following the protocol Re et al. [21] laid out. A 14 millilitre ABTS solution was combined with 5 millilitres of a 4.9 millilitre potassium persulfate ($K_2S_2O_8$) solution to create the ABTS cation radical. The resultant mixture was diluted with ethanol, after which it was left in the dark for 16 hours. The absorbance of the mxture of the plant at different concentrations and th 1 millilitre of ABTS solution was measured at 734 nm. Each measurement was calibrated using ethanol blanks, and measurements were conducted at least six minutes later. The calculation for determining the inhibition percentage of ABTS radical scavenging activity is as follows:

$$\%\text{Inhibition} = \frac{Abs.Contol - Abs.Sampl}{Abs.Conntrol} \times 100\%$$

## 2.5. Estimation of 1,1-diphenyl-2-picrylhydrazyl (DPPH)-2,2-diphenyl-1-picrylhydrazyl (DPPH) scavenging activity

The investigation was conducted using the procedures described by Blois et al. in [22]. Different concentrations of the sample in ethanol were prepared at 100–500 mg/mL. Next, the mixture was mixed with a suspension of DPPH (0.2 millimolar in ethanol) in a 1:1 ratio.

Following a 30-minute reaction, the solution's absorbance was taken at 517nm. Through the comparison of the absorbance of each sample to that of a blank solution, the scavenging activity was determined by employing the equation below.

$$\%Inhibition = \frac{Abs.Contol - Abs.Sample}{Abs.Control} \times 100\%$$

## 2.6. Estimation of hydrogen peroxide ($H_2O_2$) scavenging activity

Using the procedure outlined by Ruch et al. [23], the $H_2O_2$ scavenging capacity was determined. A 43 nm $H_2O_2$ solution was made in 0.1 m phosphate buffer (7.4). Various concentrations of the extract were mixed with a phosphate buffer and added to a solution of $H_2O_2$. The absorbance of the mixtures was taken at 230 nm, and the $H_2O_2$ scavenging activity was computed using the equation below:

$$\%Inhibition = \frac{Abs.Contol - Abs.Sample}{Abs.Control} \times 100\%$$

## 2.7. Estimation of nitric oxide (NO) scavenging activity

The procedure outlined by Tonisi [24] was used to evaluate the suppression of NO radical generation in vitro. After the addition of 2.0 ml of sodium nitroprusside, 0.5 ml of phosphate-buffer saline and 0.5 ml of leaf samples (50 mg). Incubating the mixture for 30 minutes at 25 °C, the reaction was initiated. Then, for a further thirty minutes of incubation, 0.5 millilitre of Griess reagent was added. The control for the analysis was freshly prepared in test tubes and both the absorbance of the sample and control taken at 546nm. The reagent was used as blank, and the percentage inhibition was calculated. Exploring the activity of scavenging nitric oxide:

$$\%Inhibition = \frac{Abs.Contol - Abs.Sample}{Abs.Control} \times 100\%$$

## 2.8. Anti-inflammatory activity

The plants' ability to reduce inflammation was evaluated in vitro via a technique for stabilising red blood cell membranes that was published by Anosike et al. [25]. Fresh 100 millilitres of whole mammalian blood were drawn and put into centrifuge tubes that had been heparinized. The blood was cleansed with an equivalent volume of normal saline after each of the three centrifugations, which took place at 3000 rpm for ten minutes. The blood volume was estimated, and regular saline was used to reconstitute it into a 10% V/V suspension.

## 2.9. Heat-induced haemolysis

We evaluated heat-induced hemolysis using **Juvekar** et al. [26] methodologies. The mixture was made up of 2ml: 1ml of the plant extract (100–500 µg/ml) and 1ml of a 10% red blood cell suspension obtained from Albino wistar rats. More over, The animal handling protocol was in line with the guidelines of the National Institute of Health (NIH) publication (1985) for laboratory animal care and use. Moreover, ethical approval for the use of animals in this study was obtained from the Faculty of Basic Medical Sciences Animal Research Ethics Committee (FAREC- FBMS, Approval number: 032BCH3319), University of Calabar, Nigeria. This control tube used saline as a test sample and aspirin as a reference drug. The mixture

was incubated in all centrifuge tubes at a temperature of 56 degree celcius for thirty minutes. Subsequently, the tubes were cooled using running tap water. A 560 nanometer absorbance was recorded in the supernatant above the sediment after 5 minutes of centrifugation at 2500 RPM. The test sampleswere run in triplicate. Calculating hemolysis % inhibition was as follows:

$$\%inhibition = \frac{Abs\ of\ control - Abs\ of\ sample}{Abs\ of\ control} \times 100$$

## 2.10. Mineral analysis

After digesting dried powdered samples with nitric acid and perchloric acid, minerals were extracted from the resulting filtrate. The levels of sodium (Na), potassium (K), calcium (Ca), magnesium (Mg), phosphorus (P), iron (Fe), copper (Cu), and manganese (Mn) were measured. Potassium (K) and sodium (Na) levels were assessed using the Flame photometric technique, while levels of iron (Fe), copper (Cu), manganese (Mn), calcium (Ca), and magnesium (Mg) were estimated using atomic absorption spectrophotometry, following the methods outlined in [27].

## 2.11. Proximate analysis

Analysing the proximate composition of *T. conophorum* nut, *T. cattapa* kernel, and P. americana seed was conducted by adopting the recommended protocols of analysis given by [28].

## 2.12. Determination of moisture content

The seeds of the plants were finely ground into a homogeneous powder and meticulously mixed. Triplicate aliquots of each 2-gram sample were placed on moisture dishes. The dishes were then subjected to a drying process for 24 hours at 105°C in a Memmert U.27 oven. After the drying period, the samples were left in the dessicator to get cold. The moisture content was subsequently estimatedby the following procedure:

$$\% \text{ Moisture} = \frac{(Y - Z)}{X} X\ 100$$

Consider X as the initial weight of the sample, Y as the weight of the dish plus the sample before drying, and Z as the weight of the dish plus the sample after drying. The difference between Y and Z represents a reduction in the weight of the sample resulting from the procedure of drying.

## 2.13. Determination of Ash Content

A 5 g sample of finely powdered leaf material was subjected to oven-drying at 105 °C for 3 hours using a pre-weighed ceramic crucible. Subsequently, the sample was incinerated on an electric hot plate within a fume hood to eliminate all carbonaceous matter, resulting in a greyish-white residue. This process involved an initial drying phase at 100 °C for three hours followed by an ashing phase at 500 °C for one hour in a muffle furnace. Water was added to the resultant ash to facilitate complete combustion and to verify the presence of any remaining carbon. The ash content(%) was estimated using the appropriate formulae.

$$\% \text{ Ash} = \frac{(Y - Z)}{X} X\ 100$$

Consider,

The initial sample weight is X.

The dish and its contents after ashing weigh Y.

Z represents the empty dish's weight.

## 2.14. Determination of Crude Fat Content

The 5 grams of plant extract were placed in filter paper and put in a thimble. The thimble with the plant extract were then thawed in a 50 mL jar and allowed to dry at 105 °C for 6 hours. Following drying, the thimble was placed in a Soxhlet extractor. The Soxhlet apparatus, containing a beaker, collected the extract after three washings with ethyl ether. For the sample removal from the thimble, ethyl ether was allowed to condense steadily over several hours. The extracted fat was then carefully removed from the jar to an evaporation dish of known weight. This transfer was followed by multiple ethyl ether rinses and drying under a fume hood with the blower on. The dish and its contents were dried for 30 minutes in a 105 °C oven. After drying, the weight of the dish with the oil was recorded. The crude fat percentage was then stimated from the obtained measurements.

$$\%\text{Crude fat} = \frac{W2 - W1}{Ws} \times 100\%$$

were:

W1 = empty flask weight (g).

W2 = flask and fat weight (g).

S = pre-dry sample weight.

## 2.15. Determination of crude fibre content

A 1-litre conical jar was charged with 5 grams of a finely pulverized sample. To this, 150 mL of 0.128 M H2SO4, pre-heated, was carefully added. The mixture was then boiled for 30 minutes. Following the boiling, the contents were transferred through a bent tube, ensuring thorough rinsing with hot water three times to remove any remaining material. Next, the mixture was combined with 150 mL of pre-warmed 0.15 M KOH and brought to a boil again. An antifoaming agent was added to the mixture and boiled gently for more 30 minutes. Then, the mixture was filtered, and the residue was washed with hot water and acetone thrice.

The leftover residues were ovendried at 130°C for one hour and then burnt to ash at 500°C for 30 minutes. The ash was allowed to cool and then weighed and used to determine the crude fiber cotent with the formula below:

$$\%\text{Crude fiber} = \frac{(W1 - W2)}{Ws} \times 100\%$$

were:

WS = Weight of sample prior to drying

W2 = Residue weight upon drying.

W3 = The residue's weight following ashing

## 2.16. Determination of Protein Content

According to **Sáez-Plaza et al** [29], the Kjeldahl method was employed to quantify protein content in plant extracts. In this procedure, a thirty-millilitre Gerhardt Kjeldahl digestion flask was charged with two grams of oven-dried plant material, one gram of catalyst mixture,

and fifteen millilitres of concentrated sulfuric acid. The content in the flask was then digested using heat in a fume hood until a greenish-clear solution was obtained. After a thirty-minute settling period, the solution was boiled for an additional 30 minutes before being allowed to cool. To prevent caking, ten millilitres of distilled water were added. The heated extratcs was then moved to the Kjeldahl apparatus, which was carefully assembled and calibrated. A 50-millilitre receiving testtube with 5 millilitres of boric acid indicator solution was prepared. The condenser tip was gently immersed 2 centimeters into the indicator solution. Following this, 10 millilitres of 40% sodium hydroxide were added to the processed sample within the apparatus. Distillation was initiated by opening the steam jet inlet and closing the steam bypass. Distillation was continued until the receiving flask collected 35 millilitres of distillate, at which point the inlet stopcock was closed and the steam bypass opened. The condensed distillate was titrated with 0.1N sodium hydroxide until a crimson endpoint was reached, indicating the presence of excess acid. The crude protein content was then calculated based on the titration results.

$$\% \ crude \ protein = \frac{Titre \ \times Normality \ of \ the \ acid \times 100 \times 6.25}{1000 \times weight \ of \ sample}$$

## 2.17. Determination of carbohydrates

The carbohydrate content (%) of the plant extract was estimated by adding the percentages of moisture, ash, crude protein, and fat, then subtracting this total from 100%, following the method outlined by Ijarotimi et al. [30].

## 2.18. Vitamin analysis

The High-Performance Liquid Chromatography (HPLC) technique was employed to analyse vitamin concentration in the plant extracts. The mobile phase for analyzing water-soluble vitamins consisted of 1.03 g of hexane, sulfuric acid, sodium salt, and 6.8 g of potassium dihydrogen phosphate dissolved in 940 ml of HPLC-grade water. To this solution, 5 ml of trimethylamine was added, and the pH was adjusted to 3.0 using orthophosphoric acid. HPLC-grade acetonitrile was used as mobile phase B. [31]

For fat-soluble vitamins estimation, the saponification process was employed [32]. A plant sample was saponified using 5 g of ethanolic potassium hydroxide. Additionally, another 5 g of the sample underwent alkaline hydrolysis using the same ethanolic potassium hydroxide. The mixture was refluxed for approximately 45 minutes. The saponified digest was then extracted with a 1:1 mixture of hexane and diethyl ether, using a volume of 0.1 ml. A 10 ml portion of this organic layer was repeatedly washed with water, then dried by evaporation. The residue was reconstituted with 200 μl of mobile phase B and subsequently injected into the HPLC system for analysis.

## 2.19. Analysis of phytochemicals

The phytochemical profile of the plant extract was characterized using gas chromatography-mass spectrometry (GC-MS) [15]. The analysis was performed on an Agilent 8860 GC system coupled with a mass spectrometer detector. A fused silica capillary column coated with DB-5MS (0.25 mm internal diameter, 0.15 μm film thickness) was utilized, and samples were injected in pulsed splitless mode. Helium served as the carrier gas, while a mixture of hydrogen and compressed air was used as the ignition gas. The column head pressure was maintained at 20 psi, ensuring a constant flow rate of 1 mL/min. The column oven temperature was initially set at 55°C for 0.4 minutes, then programmed to increase to 200°C at a rate of 25°C/

min. This was followed by a further temperature ramp to 280°C at 8°C/min, and finally to 300°C at 25°C/min, where it was held for 2 minutes. Compound identification was achieved through retention time analysis, with earlier elution corresponding to components with lower retention times. For the sample preparation, a 50 mg aliquot of the plant extract was dissolved in a 10 mL mixture of hexane and dichloromethane in a 1:1 ratio and then transferred to a beaker for analysis.

## 2.20. Molecular docking analysis

A total of forty-five bioactive constituents were identified in the seeds of P. americana, the nuts of *Treculia conophorum*, and the kernels of *T. catappa*. *P. americana* contained 13 compounds, Treculia conophorum had 19 compounds, and *T. catappa* included 13 compounds. These compounds, along with acarbose, were used as ligands in the docking study. The bioactive compounds extracted from *P. americana* seeds and *T. catappa* kernels were present in significant proportions, with thirteen compounds identified in each. Additionally, nineteen bioactive compounds were extracted from Treculia conophorum nuts. The molecular structures of these compounds were retrieved from the PubChem compound database (https://pubchem.ncbi.nlm.nih.gov). The three-dimensional structures of alpha-amylase and alpha-glucosidase were retrieved from the Protein Data Bank (www.rcsb.org). After downloading the SDF files containing the structural data of the ligands, molecular docking was performed with alpha-amylase and alpha-glucosidase as targets. Chimera 1.14 was used to remove unnecessary water molecules and non-standard residues, while incorporating hydrogen atoms and charges. The ligands were prepared and converted into SDF formats using the PyRx program. Energy minimization and force field optimization were conducted on the ligands.

With the use of Autodock Vina in PyRx, the molecular docking of ligands to the protein targets was performed, allowing the assessment of their binding affinities. Chimera 1.14 and Discovery Studio 2020 were employed to analyze the interactions between the proteins and ligands.

## 2.21 Statistical analysis

Data analysis was conducted using GraphPad Prism version 8.21 (GraphPad Software, La Jolla, CA, USA). Statistical significance was determined via one-way analysis of variance (ANOVA), followed by a post hoc Tukey's test, with a significance threshold set at $P < 0.05$. Results are expressed as the mean ± standard error of the mean (SEM) for each dataset, with each condition measured in triplicate ($n = 3$).

## 3. Results

The *in vitro* inhibitory activity of extracts from *Persea americana*, *T. conophorum*, and *T. catappa* against α-glucosidase is detailed in Table 1. Among the tested extracts, *P. americana* exhibited the most potent inhibition of α-glucosidase, with an $IC_{50}$ value of 208 µg/mL. This was followed by *T. catappa* extract, which demonstrated an $IC_{50}$ value of 236 µg/mL. Notably, this substantial inhibitory effect was recorded at a concentration of 150 mg/mL for both extracts.

The *in vitro* inhibitory effects of extracts from *P. americana*, *T. conophorum*, and *T. catappa* on α-amylase activity are detailed in Table 2. The data indicate that *P. americana* extract demonstrated the highest inhibition of α-amylase activity, with an $IC_{50}$ value of 312 µg/mL, in comparison to *T. conophorum* and *T. catappa* extracts, where *T. conophorum* exhibited the lowest inhibitory activity.

**Table 1.  α-Glucosidase Inhibitory activity of *P. americana* seed, *T. conopherum* nut and *T. catappa* kernel.**

| Samples | Concentration (mg/ml) | | | | | IC 50 (µg/mg) |
|---|---|---|---|---|---|---|
| | 10 | 20 | 50 | 100 | 150 | |
| Acarbose | 8.88 ± 0.02 | 9.16 ± 0.01 | 9.16 ± 0.01 | 13.35 ± 0.02 | 15.71 ± 0.03 | -245.27 |
| p. Americana | 8.93 ± 0.026 | 15.68 ± 0.023[a] | 20.93 ± 0.029[a] | 29.15 ±0.056 | 38.09 ± 0.038 [a] | 208.95 |
| T. Conopherum | 28.55 ± 0.043[a] | 7.47 ± 0.032 [b] | 8.08 ± 0.032[b] | 32.54 ± 0.080 [a] | 33.11 ± 0.055 | 278.03 |
| T. Catappa Kernel | -9.29 ± 0.012[b] | 8.27 ± 0.023 | 20.93 ± 0.023 | 22.26 ± 0.026 [b] | 27.84 ± 0.091[b] | 236.28 |

Mean Values having different letters as superscripts across the row and down the column are considered significant at P < 0.05.

Letter 'a' as superscript means significantly higher (P < 0.05) compared to the standard value while letter 'b' as the superscript means significantly lower (P > 0.05) compared to the standard.

**Table 2.  α-amylase inhibitory activity *P. americana* seed, *T.conopherum* nut and *T. catappa* kernel.**

| Samples | Concentration (mg/ml) | | | | | IC 50 (µg/mg) |
|---|---|---|---|---|---|---|
| | 10 | 20 | 50 | 100 | 150 | |
| Acarbose | 55.31 ± 0.66[a] | 66.37 ± 0.38[a] | 74.80 ± 1.07[a] | 86.35 ± 0.64[a] | 91.93 ± 0.55[a] | -37.86 |
| P. americana | 0.07 ± 0.006 | 7.78 ± 0.026 | 11.56 ± 0.043[a] | 15.55 ± 0.044 | 25.33 ± 0.023[a] | 312.68 |
| T. Conopherum | 3.86 ± 0.023 | 6.06 ± 0.023[b] | 5.73 ± 0.030[b] | 8.08 ± 0.038[b] | 13.13 ± 0.034[b] | 806.10 |
| T. Catappa Kernel | -0.59 ± 0.026[b] | 8.53 ± 0.078 | 9.27 ± 0.038 | 15.78 ± 0.059[a] | 18.86 ± 0.064 | 402.97 |

Mean Values having different letters as superscripts across the row and down the column are considered significant at P < 0.05. Letter 'a' as superscript means significantly higher (P < 0.05) compared to the standard value while letter 'b' as the superscript means significantly lower (P > 0.05) compared to the standard.

Table 3 presents the antioxidant capacities of these extracts, measured by their scavenging effect on the ABTS radical. The *T. conophorum* extract displayed the most potent scavenging activity, with an $IC_{50}$ value of 976.21 µg/mL, whereas the *P. americana* extract had the least efficacy, with an $IC_{50}$ value of 5190.67 µg/mL. This suggests that *T. conophorum* extract possesses a significant capability to scavenge ABTS radicals.

The DPPH radical scavenging activity of the three extracts is detailed in Table 4. The *T. cattapa* extract demonstrated the most potent DPPH radical scavenging activity, with an IC50 value of 941.95 µg/mL. This was followed by the *T. conophorum* extract, which had an IC50

**Table 3.  ABTS Radical Scavenging Activity *P.americana* seed, *T.conopherum* nut, and *T. catappa* kernel.**

| Samples | Concentration (mg/ml) | | | | | IC 50 (µg/ml) |
|---|---|---|---|---|---|---|
| | 10 | 20 | 50 | 100 | 150 | |
| Ascorbic acid | 96.98 ± 0.01 | 98.35 ± 0.01 | 96.98 ± 0.01 | 97.58 ± 0.01 | 98.35 ± 0.01 | -8924.34 |
| P. americana | 4.85 ± 0.029 | 5.79 ± 0.023 | 4.80 ± 0.021 | 4.39 ± 0.018 | 4.15 ± 0.041 | 5190.67[b] |
| T. Conopherum | 3.94 ± 0.018 | 5.09 ± 0.032 [b] | 9.82 ± 0.058 | 9.39 ± 0.036 | 11.06 ± 0.029 | 976.21[a] |
| T. Catappa Kernel | 5.51 ± 0.020 | 6.08 ± 0.034 | 6.17 ± 0.041 | 7.14 ± 0.038 | 7.44 ± 0.064 | 3363.45 |

Mean Values having different letters as superscripts across the row and down the column are considered significant at P < 0.05. Letter '

a' as superscript means significantly higher (P < 0.05) compared to the standard value while letter '

b' as the superscript means significantly lower (P > 0.05) compared to the standard.

**Table 4. DPPH Scavenging Activity of *P.americana* seed, *T.conopherum* nut, and *T. catappa* kernel.**

| Samples | Concentration (mg/ml) | | | | | IC 50 (µg/mg) |
|---|---|---|---|---|---|---|
| | 10 | 20 | 50 | 100 | 150 | |
| Ascorbic acid | 96.98 ± 0.01 | 98.35 ± 0.01 | 96.98 ± 0.01 | 97.58 ± 0.01 | 98.35 ± 0.01 | -8924.34 |
| *P. Americana* | 25.39 ± 0.012 | 25.42 ± 0.017 | 27.35 ± 0.015 | 27.86 ± 0.024 | 28.09 ± 0.020 | 1417.35[b] |
| *T. Conopherum* | 23.51 ± 0.015 | 25.23 ± 0.020 | 27.12 ± 0.064 | 26.66 ± 0.023 | 27.08 ± 0.036 | 1288.43 |
| *T. Catappa Kernel* | 28.53 ± 0.047 | 29.22 ± 0.041 | 30.14 ± 0.075 | 31.37 ± 0.050 | 31.75 ± 0.050 | 941.95[a] |

Mean Values having different letters as superscripts across the row and down the column are considered significant at P < 0.05. Letter 'a' as superscript means significantly higher (P < 0.05) compared to the standard value while letter 'b' as the superscript means significantly lower (P > 0.05) compared to the standard.

value of 1288.43 µg/mL. The *P. americana* extract exhibited the lowest DPPH scavenging activity, with an IC50 value of 1417.85 µg/mL.

The hydrogen peroxide neutralizing capacity of the extracts, as presented in Table 5, demonstrated a concentration-dependent response. The *T. cattapa* extract exhibited the highest neutralizing activity, with an IC50 value of 265.22 µg/mL. This was followed by the *P. americana* extract, which had an IC50 value of 495.00 µg/mL. In contrast, the *T conophorum* extract showed the least activity, with an IC50 value of 637.61 µg/mL.

**Table 5. $H_2O_2$ Scavenging Activity of *P. americana* seed, *T. conopherum* nut and *T.catappa* kernel.**

| Samples | Concentration (mg/ml) | | | | | IC 50 (µg/mg) |
|---|---|---|---|---|---|---|
| | 10 | 20 | 50 | 100 | 150 | |
| Ascorbic acid | 86.74 ± 0.01 | 90.3 ± 0.010 | 91.82 ± 0.01 | 95.24 ± 0.01 | 98.51 ± 0.01 | -708.57 |
| *P. americana* | 23.55 ± 0.055 | 26.73 ± 0.055 | 29.82 ± 0.055 | 31.05 ± 0.012 | 31.60 ± 0.085 | 495 |
| *T. Conopherum* | 29.68 ± 0.056 | 31.74 ± 0.055 | 32.83 ± 0.029 | 33.89 ± 0.104 | 34.90 ± 0.029 | 637.61[b] |
| *T. Catappa Kernel* | 4.96 ± 0.062 | 28.84 ± 0.071 | 30.30 ± 0.078 | 32.31 ± 0.096 | 31.72 ± 0.107 | 265.22[a] |

Mean Values having different letters as superscripts across the row and down the column are considered significant at P < 0.05. Letter 'a' as superscript means significantly higher (P < 0.05) compared to the standard value while letter 'b' as the superscript means significantly lower (P > 0.05) compared to the standard.

**Table 6. NO Scavenging Activity of *P.americana* seed, *T.conopherum* nut, and *T. catappa* kernel.**

| Samples | Concentration (mg/ml) | | | | | IC 50 (µg/mg) |
|---|---|---|---|---|---|---|
| | 10 | 20 | 50 | 100 | 150 | |
| Ascorbic acid | 68.18 ± 0.01 | 72.63 ± 0.01 | 80.24 ± 0.01 | 86.94 ± 0.01 | 91.36 ± 0.01 | -121.380 |
| *P. americana* | 25.36 ± 0.047 | 26.41 ± 0.044 | 27.29 ± 0.080 | 27.83 ± 0.042 | 28.12 ± 0.018 | 1403.14 |
| *T. Conopherum* | 23.61 ± 0.050 | 25.29 ± 0.041 | 27.11 ± 0.055 | 26.10 ± 0.055 [b] | 27.07 ± 0.023 | 1438.92[b] |
| *T. Catappa Kernel* | 28.88 ± 0.032 | 29.27 ± 0.023 | 30.16 ± 0.074 | 31.34 ± 0.032 | 31.77 ± 0.045 | 956.94[a] |

Mean Values having different letters as superscripts across the row and down the column are considered significant at P < 0.05.

Letter 'a' as superscript means significantly higher (P < 0.05) compared to the standard value while letter 'b' as the superscript means significantly lower (P > 0.05) compared to the standard.

Table 6 presents the outcomes of the nitric oxide scavenging assay for the three extracts. Notably, the *T. cattapa* extract demonstrated the most potent NO scavenging activity, exhibiting an IC50 of 956.94. Following this, the *P. americana* extract displayed a moderately lower activity with an IC50 value of 1403.14. Conversely, the *T. conophorum* extract exhibited the least activity, with an IC50 value of 1438.92. Moreover, the scavenging efficacy against NO increased proportionally with extract concentration.

The findings from the human red blood cell membrane stabilization test, as depicted in Fig 1, reveal the anti-inflammatory efficacy. Extracts from *T. cattapa*, *P. americana*, and *T. conophorum* demonstrated a concentration-dependent inhibition of inflammation, with the protective efficacy escalating alongside the concentration of the samples.

It was observed that *T. cattapa extract* produced a higher inhibition of RBC hemolysis with an IC50 value of 212.05 while *T. conophorum* extract had the least inhibition with an IC50 of 423.76.

The result from the mineral analysis as presented in Table 7 shows that the three samples indicated that the phosphorus concentration was significantly higher in *T. catappa* kernel and *T. conophorum* when compared with *P. americana*. All the three extracts recorded zero iron (Fe) content while *P. americana* had significantly higher concentrations of copper, manganese, potassium and calcium with a mean concentration of 0.42, 0.64, 0.04 and 0.93 mg/g respectively when compared with *T. cattapa* and *T. conophorum*. *T. cattapa* had a higher

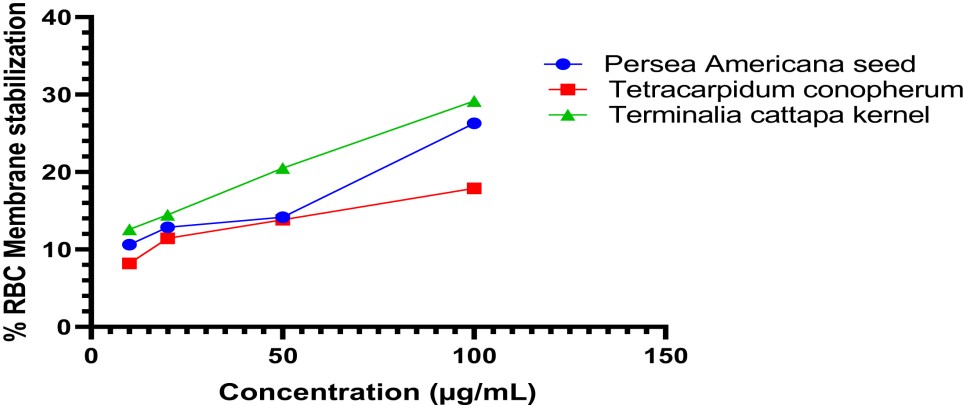

**Fig 1.** *In-vitro* anti-inflammatory activity of extracts of *Terminalia cattapa* Seed, *Terminalia cattapa* Seed and *Tetracarpidium conophorum* nut.

**Table 7.** Mineral composition of *P. Americana, T. conophorum* and *T. cattapa*.

| Samples | Fe (mg/g) | P.(mg/g) | Cu (mg/g) | Mn (mg/g) | Mg (mg/g) | K (mg/g) | Na (mg/g) | Ca (mg/g) |
|---|---|---|---|---|---|---|---|---|
| *Persea Americana* | | 4.70 ± 0.10[b] | 0.42 ± 0.01 [a] | 0.643 ± 0.006 [a] | 0.863 ± 0.006[a] | 2.110 ± 0.010[a] | 0.043 ± 0.006[b] | 0.93 ± 0.02[a] |
| *Tetracarpidum Conopherum* | | 6.33 ± 0.06 | 0.34 ± 0.01 | 0.250 ± 0.010 | 0.707 ± 0.006[b] | 1.607 ± 0.006[b] | 0.067 ± 0.006[a] | 0.78 ± 0.01 |
| *Terminalia Cattapa* | | 7.07 ± 0.06[a] | 0.35 ± 0.01 | 0.217 ± 0.016[b] | 0.917 ± 0.006[a] | 1.715 ± 0.006 | 0.060 ± 0.001 | 0.68 ± 0.01[b] |

Mean Values having different letters as superscripts across the row and down the column are considered significant at P < 0.05.

Letter '

a' as superscript means significantly higher (P < 0.05) compared to the standard value while letter '

b' as the superscript means significantly lower (P > 0.05) compared to the standard

**Table 8. Proximate composition of *P. Americana, T. conopherum* and *T. cattapa.***

| Sample | MC (%) | Crude Fibre (%) | CHO (%) | Protein (%) | Lipids (%) | Ash (%) |
|---|---|---|---|---|---|---|
| *Persea americana* | 51.58 ± 0.21[a] | 6.02 ± 0.02[b] | 0.55 ± 0.03[b] | 0.94 ± 0.01[b] | 3.33 ± 0.02[b] | 11.00 ± 0.01[b] |
| *Tetracarpidum conopherum* | 21.07 ± 0.04 | 12.07 ± 0.10[a] | 1.22 ± 0.02 | 1.77 ± 0.01 | 29.83 ± 0.4 | 25.01 ± 0.02 |
| *Terminalia cattapa* | 15.01 ± 0.02[b] | 8.40 ± 0.01 | 2.45 ± 0.02[a] | 2.12 ± 0.02[a] | 30.02 ± 0.05[a] | 40.65 ± 0.05[a] |

Mean Values having different letters as superscripts across the row and down the column are considered significant at P < 0.05.

Letter 'a' as superscript means significantly higher (P < 0.05) compared to the standard value while letter 'b' as the superscript means significantly lower (P > 0.05) compared to the standard.

magnesium concentration with a mean of 0.92mg/g when compared with *T. conophorum* and *P. americana*.

The result in Table 8 shows the proximate composition of *T. conophorum, T. cattapa* and *P. americana*. Comparing the three plants, the moisture content was significantly higher in *P. americana* when compared with *T. conophorum* and *T. cattapa*. Concentration of the carbohydrates, proteins, lipids and ash was significantly higher in *T. cattapa* when compared with *P. americana* and *T. conophorum* while the crude fibre concentration was significantly higher in *T. conophorum* when compared with the *T. cattapa* and *P. americana*.

Table 9 shows the variations in concentrations of water-soluble vitamins in *T. conophorum, T. cattapa* and *P. americana* seeds. The most preponderant vitamins across the three samples are vitamins B1 and B5 as shown in Table 9 above. Vitamin B3, B10 and B12 concentrations are very low across the three samples, vitamins B2, B3 and B9 are largely observed in *P. americana* seed while *T. conophorum* displayed the highest content of vitamin C.

The result in Table 10 shows the HPLC analysis of fat-soluble vitamins shows that nine (9) fat-soluble vitamins (medianone, linoleic acid, retinoic acid, vitamin K2, cholecalciferol, vitamin K11, retinol, cis-retinal, and tocopherol) were analysed and varying concentrations of the vitamins was recorded for the different samples. Medianone and linoleic acid were not detected in all three (3) samples. *T. conophorum* recorded the highest concentration of retinoic acid with a concentration of 0.487195 mg/100g followed closely by *P. americana* seed

**Table 9. Water-soluble vitamin composition of *P. Americana, T. conopherum* and *T.Cattapa.***

| Water Soluble Vitamins | | | |
|---|---|---|---|
| | *T. conopherum(mg/100g)* | *T. cattapa kernel (mg/100g)* | *P. americana(mg/100g)* |
| Vitamin B1 | 2.06 ± 0.05[a] | 1.66 ± 0.05 | 0.52 ± 0.01[b] |
| Vitamin B2 | 0.005 ± 0.0004 | 0.002 ± 0.0002[b] | 0.22 ± 0.02 [a] |
| Vitamin B3 | 1.29e-5 ± 3.61e-7 | 4.67e-6 ± 4.04e-7[a] | 0.0 ± 0.0[b] |
| Vitamin B5 | 0.34 ± 0.007[b] | 1.05 ± 0.009 [a] | 0.45 ± 0.01 |
| Vitamin B6 | 0.32 ± 0.005 | 0.0 ± 0.0[b] | 0.54 ± 0.022[a] |
| Vitamin B9 | 0.037 ± 0.002 | 0.036 ± 0.004 | 0.36 ± 0.013 [a] |
| Vitamin B10 | 0.004 ± 0.0001 [a] | 0.0 ± 0.0 | 0.0 ± 0.0 |
| Vitamin B12 | 0.004 ± 0.0001 | 0.0 ± 0.0 [b] | 0.047 ± 0.002[a] |
| Vitamin C | 1.60 ± 0.02 | 3.6e-4 ± 4.5e-5 [a] | 0.091 ± 0.005[b] |
| Total | 4.37 ± 0.08[a] | 2.75 ± 0.04 | 2.23 ± 0.02[b] |

Mean Values having different letters as superscripts across the row and down the column are considered significant at P < 0.05. Letter 'a' as superscript means significantly higher (P < 0.05) compared to the letter 'b' as the superscript means significantly lower (P > 0.05).

**Table 10. Fat-soluble vitamin composition of *T. conopherum* nut, *T.catppa* kernel and *P.***

| | Fat Soluble Vitamins | | |
| --- | --- | --- | --- |
| | *T. conopherum(mg/100g)* | *T. cattapa kernel(mg/100g)* | *P. americana(mg/100g)* |
| Medianone | 0.0 ± 0.0 | 0.0 ± 0.0 | 0.0 ± 0.0 |
| Linoleic acid | 0.0 ± 0.0 | 0.0 ± 0.0 | 0.0 ± 0.0 |
| Retinoic acid | 0.49 ± 0.014[a] | 0.29 ± 0.01[b] | 0.43 ± 0.019 |
| Vitamin K2 | 0.0 ± 0.0[b] | 1.3e-3 ± 6.5e-5 | 1.18 ± 0.03[a] |
| Cholecalciferol | 3.7e-4 ± 6e-5[a] | 1.1e-3 ± 8.5e-5 | 0.0 ± 0.0[b] |
| Vitamin K1 | 6.42 ± 0.085 [a] | 0.76 ± 0.032[b] | 1.9e-3 ± 3.1e-4 |
| Retinol | 0.0 ± 0.0 | 0.0 ± 0.0 | 0.94 ± 0.008 [a] |
| Cis Retinal | 0.069 ± 0.003 [a] | 0.0 ± 0.0 [b] | 0.005 ± 0.0003 |
| Tocopherol | 0.48 ± 0.026 [a] | 0.048 ± 0.005 | 0.037 ± 0.001 [b] |
| Total | 7.45 ± 0.075 [a] | 1.10 ± 0.023 [b] | 2.59 ± 0.05 |

Mean Values having different letters as superscripts across the row and down the column are considered significant at P < 0.05.

Letter 'a' as superscript means significantly higher (P < 0.05) compared to the letter 'b' as the superscript means significantly lower (P>).

(0.422626 mg/100g) and *T. cattapa* kernel (0.291829 mg/100g). *P. americana* seed had the highest concentration of vitamin K2 with a value of 1.145608 mg/100g followed by *T. cattapa* kernel (0.00121 mg/100g) and undetected in *T. conophorum*. *T. cattapa* kernel had the highest concentration of cholecalciferol with a value of 0.00104 mg/100g followed by *T. conophorum* kernel (0.00031 mg/100g) and undetected in *P. americana* seed. *T. conophorum* nut recorded the highest concentration of vitamin K1 among all fat-soluble vitamins with a value of 6.416094 mg/100g, followed by *T. cattapa* kernel (0.794015 mg/100g) and *P. americana* seed (0.001596 mg/100g). retinol was only detected in *P. americana* seed at a concentration of 0.934947 mg/100g. *T. conophorum* recorded the highest concentration of cis-retinal with a value of 0.069507 mg/100g followed by *Persea americana* seed (0.004863 mg/100g) and was undetected in *T. cattapa* kernel. *T. conophorum* recorded the highest concentration of tocopherol with a value of 0.479305 mg/100g followed by *T. cattapa* (0.042296 mg/100g) and avocado (0.036644 mg/100g). the total concentrations of fat-soluble vitamins for walnut, almond and avocado were recorded to be 7.45241, 1.13039 and 2.546285 mg/100g respectively. The result from the HPLC analysis of fK2, cholecalciferol, vitamin K11, retinol, cis-retinal, and tocopherol) were analysed and of 0.479305 mg/100g followed by *T. cattapa* (0.042296 mg/100g) and avocado (0.036644 mg/100g). The total concentrations of fat-soluble vitamins for walnut, almond and avocado were recorded to be 7.45241, 1.13039 and 2.546285 mg/100g respectively. Table 11, Table 12, and Table 13, shows the phytochemical composition of *P. americana* seed, *T. conophorum* nut and *T. cattapa* kernel. *P. americana* seed had thirteen compounds present, *T. conophorum* nut had sixty-five compounds with twenty-six of the compounds present in appreciable percentages, while *T. cattapa* had forty-six compounds, and only twenty compounds were present in appreciable percentages.

### 3.1 Molecular docking

The findings from the analysis of the three-dimensional (3D) and two-dimensional (2D) structures, along with docking scores, of α-glucosidase and amylase, in complex with the co-crystalized compounds acarbose, as well as compounds sourced from the seeds of Persea americana (avocado), the nut of Terminalia conophorum, and the kernel of Terminalia cattapa, are depicted in Figs 2 and 3 (A–D, respectively), and Tables 14 and 15. The results revealed that compounds extracted from

**Table 11. The phytochemical composition of *P. americana* seed.**

| Peak | Retention Time | Area | Height | Compound Name |
|------|------|------|------|------|
| 1. | 0.191 | 298129 | 292671 | Methylene Chloride |
| 2. | 0.266 | 10101 | 13064 | Hexane, 2,2,3 – Trimethyl |
| 3. | 0.313 | 38114 | 28187 | 1 – Hexene, 3, 4 – dimethyl |
| 4. | 0.434 | 1.039960 | 467930 | 2 – [2 – (2 – Chloro – ethoxy) – ethoxy] Phenol |
| 5. | 0.539 | 117116 | 98643 | 2 [2 – (2 – Chloro – ethoxy) – ethoxy] Phenol |
| 6. | 0.649 | 828590 | 854233 | Propanoic acid, 2–Chloro methylester |
| 7. | 0.915 | 27271 | 17941 | 2 – Heptane |
| 8. | 0.969 | 23778 | 9320 | Propanedinitrile |
| 9. | 23.068 | 11883 | 5333 | 3 – Pyrrolidinol |
| 10. | 23. 378 | 33703 | 15084 | 3 – Butnenitrile |
| 11. | 23.854 | 7478 | 3979 | Succinimide |
| 12. | 23.889 | 71731 | 4869 | 3 – oxo – hexanedoic acid |
| 13. | 24.113 | 6512 | 4249 | Pent – 2 – ynal |

**Table 12. Showing phytochemical composition of *Tetracarpidium conophorum* nut.**

| PEAK | RETENTION TIME | AREA | HEIGHT | COMPOUND NAME |
|------|------|------|------|------|
| 1. | 0.478 | 26811089 | 1228244 | 1 – Chloro – 2 – methylbutane |
| 3. | 0.544 | 141976131 | 23613671 | 2 – methylpentane |
| 4. | 0.631 | 361988903 | 30886391 | 1, 2, 3, - Trimethyldiaziridine |
| 5. | 0.670 | 498406604 | 28383724 | n – Hexane |
| 6. | 0.716 | 629855435 | 22785926 | n – Hexane |
| 7. | 0.788 | 527102168 | 40430712 | 2 – Pentene |
| 8. | 0.833 | 994436358 | 39238102 | 1 – Pentene |
| 9. | 0.967 | 1638554718 | 56477054 | 1, 5 – Hexadiyne |
| 10. | 1.007 | 2307053886 | 67234010 | 1,3 – Hexadien – 5 – yne |
| 11. | 1.082 | 1444164395 | 49369546 | Cycloheptane |
| 12. | 1.148 | 1766577507 | 50150077 | Cycloheptane |
| 13. | 1.243 | 951774820 | 33567219 | Hexane, 2, 3, 4 – Trimethyl |
| 14. | 1.451 | 4756842379 | 65480191 | 2 – Hexenal |
| 15. | 1.497 | 145223817 | 13438932 | Cyclop enthane, 1, 2, 3 – trimethyl |
| 16 | 1.539 | 128345126 | 10884902 | Cyclop enthane, 1, 2, 3 – trimethyl |
| 17 | 1.603 | 50277716 | 3493148 | Cyclop enthane, 1, 2, 3 – trimethyl |
| 18 | 1.694 | 5300854 | 1986273 | Hexane, 3 – ethyl – 4 methyl |
| 19 | 1.817 | 2438438366 | 56473048 | Toluene |
| 20 | 1.888 | 2036085476 | 70177423 | Toluene |
| 21 | 1.951 | 1444322840 | 52363410 | Cyclohexane, 1, 4 – dimethyl - |
| 22 | 2.015 | 89222315 | 5663810 | Cyclohexane, 1, – ethyl – 3 methyl = |
| 23 | 2.161 | 1664624894 | 56658079 | Aziridine, 2, 2 – dimethyl |
| 24 | 2.224 | 81601194 | 9440866 | Cyclohexane, 1, 4 – dimethyl -, trans - |
| 25 | 2.589 | 379700427 | 21230980 | Cyclohexane, ethyl - |
| 26 | 3.212 | 971756114 | 30465043 | O – xylene |
| 27 | 3.557 | 212352590 | 12710509 | Nonane |

**Table 13. Showing phytochemical composition of *Terminalia catappa* kernel.**

| PEAK | RETENTION TIME | AREA | HEIGHT | COMPOUND NAME |
|---|---|---|---|---|
| 1. | 0.514 | 36392063 | 1850002 | Decane, 2, 2, 4 – trimethyl- |
| 2. | 0.555 | 26051968 | 1606216 | 1 – tridecenyl propionate |
| 3. | 0.605 | 76742383 | 6408320 | Azetidine, 1, 2 – dimethyl |
| 4. | 0.631 | 149728861 | 8609733 | Penthane, 3 – methyl |
| 5. | 0.669 | 256574145 | 9433110 | 1 – Ethylcyclopropand |
| 6. | 0.765 | 432368594 | 11767719 | 1 – Octane |
| 7. | 0.934 | 788862064 | 14618245 | 1 – methylbutyl |
| 8. | 1.014 | 333185452 | 12326078 | 1 – pentanol |
| 9. | 1.063 | 680061191 | 12655380 | Cyclohexane |
| 10. | 1.268 | 1553543647 | 15886171 | 1 – ethyl – 3 methyl |
| 11. | 1.581 | 35155477 | 807586 | Carbonic acid |
| 12. | 1.760 | 827049282 | 10960509 | Cyclohexane, 1, 3 – dimethyl |
| 13. | 1.988 | 293179015 | 6654154 | 1H – pyrazol – 4 – amine |
| 14. | 2.055 | 135465481 | 2590241 | 1, 2 – dimethyl – (cis/trans) |
| 15. | 2.466 | 58919201 | 578014 | 3, 4 – dimethyl – 2 – hexane |
| 16. | 2.665 | 15581328 | 215384 | 2, 3, - dimethyl – 3 –heptane |
| 17. | 3.064 | 57505534 | 579069 | P – xylene |
| 18. | 3.241 | 43258184 | 489148 | O – xylene |
| 19. | 3.437 | 30465674 | 461560 | P – xylene, Benzene |
| 20. | 21.823 | 282577859 | 3402923 | 9, 12, 15 – octade carien – 1 - ol |

P. americana seed, T. conophorum nut, and T. cattapa kernel exhibit variable degrees of binding affinities towards α-glucosidase and amylase, as inferred from the changes in Gibbs free energy (ΔG). In terms of α-glucosidase inhibition, four compounds—Pseudococaine (-6.6 kcal/mol), M-(1-methylbutyl) phenylmethylcarbamate (-6.5 kcal/mol) from T. cattapa, and o-xylene (-5.4 kcal/mol) from T. conophorum—demonstrated binding affinities closely resembling that of the standard drug acarbose (-7.8 kcal/mol), and surpassing the affinity of the co-crystallized inhibitor ligand 1-deoxynojirimycin (-5.3 kcal/mol). Similarly, against α-amylase, compounds such as dimethyl phthalate (-6.0 kcal/mol), Pseudococaine (-7.0 kcal/mol), and M-(1-Methylbutyl)phenyl methylcarbamate (-6.2 kcal/mol) from T. cattapa, along with Methyl 2-chloro-5-nitrobenzoate (-5.8 kcal/mol) and 2-chloro-3-oxohexanedioic acid (-5.4 kcal/mol) from P. americana, exhibited docking scores comparable to the standard drug acarbose (-8.4 kcal/mol), yet displaying greater affinity compared to the co-crystallized inhibitor ligand nitrate (-3.4 kcal/mol).

## 3. Discusion

Given the seriousness of diabetes and its associated problems, it is crucial to have access to new treatment advancements and affordable medication in middle- and low-income nations. This is essential for effectively managing the harmful consequences of this metabolic illness [33]. Although there is a wide range of medications to manage hyperglycemia and diabetes-related problems, there are still worries over the potential adverse effects. Given the seriousness of diabetes and its associated problems, the accessibility and affordability of new therapeutic advancements in medication are crucial in middle- and low-income countries[33]. Therefore, it is crucial to investigate alternative antidiabetic substances, particularly those derived from natural sources, in order to improve the management of diabetes. This is particularly relevant due to the vast array of metabolites found in plants [34, 35].

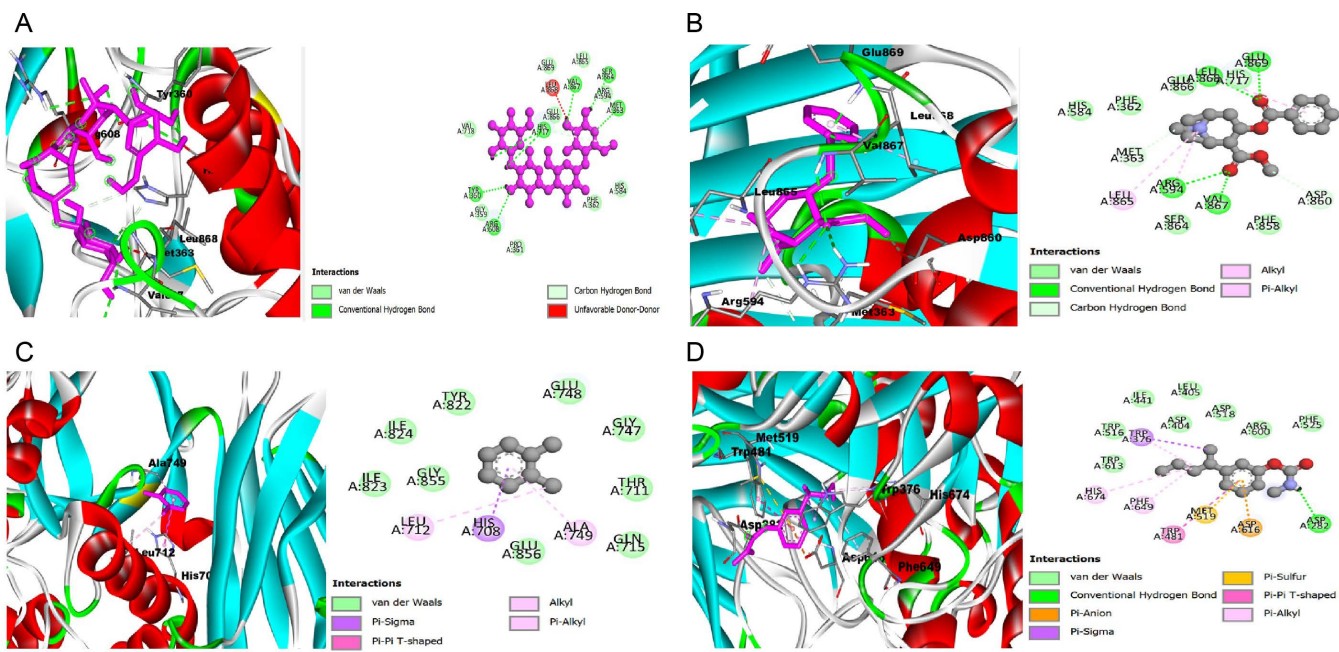

**Fig 2. The three and two-dimensional (3D and 2D) view of the molecular interactions of amino acid residues of α-glucoside with Acarbose(A), Pseudococaine (B), O-xylene (C), M-(1-Methylbutyl)phenyl methylcarbamate (D).**

This research study revealed that extracts from the seeds of Persea americana, the nuts of Tetracarpidium conophorum, and the kernels of Terminalia cattapa effectively inhibit the enzymes alpha amylase and glucosidase. This suggests that these extracts have the potential to be used as treatments for managing diabetic mellitus. Alpha-glucosidase is an enzyme found in the brush border of the epithelium of the small intestine. It specifically operates on $\alpha$ $(1\to4)$ linkages. It facilitates the hydrolysis of disaccharides and starch into glucose. Glucosidase inhibitors decrease the speed at which carbohydrates are broken down and slow down the process of absorbing carbohydrates from the digestive system [36]. Alpha amylase is a key enzyme in humans that plays a direct role in the decomposition of starch into easily digestible sugars. It breaks down large polysaccharides into smaller oligosaccharides and disaccharides. These smaller molecules are then broken down into monosaccharides. The monosaccharides are subsequently absorbed via the small intestine into the hepatic portal vein, leading to a rise in glucose levels after a meal. The enzymes exhibit an inhibitory mechanism that involves the delay of carbohydrates and the reduction of glucose absorption rate [37, 38], and our r findings align with the study conducted by Etassala et al. [39].

The use of antioxidants in medical therapy has recently become more significant, particularly in the context of diabetes [40]. Experimental studies have shown that antioxidants can significantly decrease the occurrence of problems associated with diabetes. The study found that the *T. cattapa* kernel had greater levels of antioxidants, such as ABTS, DPPH, nitric oxide, and hydrogen peroxide scavenging activity, compared to the *P. americana* seed and *T. conophorum* nut. This suggests that the plants may have a considerable scavenging activity. Our findings are consistent with the findings reported by Udeozor et al. [41].

In animals, inflammation is a defensive reaction to cell damage and plays a key role in the progression of problems in diabetes mellitus. This process is mostly associated with the involvement of innate immunity [1]. The initiation of type 1 diabetes is mostly caused by infectious viral or autoimmune mechanisms, as stated by Lemos et al., and Zajec et al [42, 43].

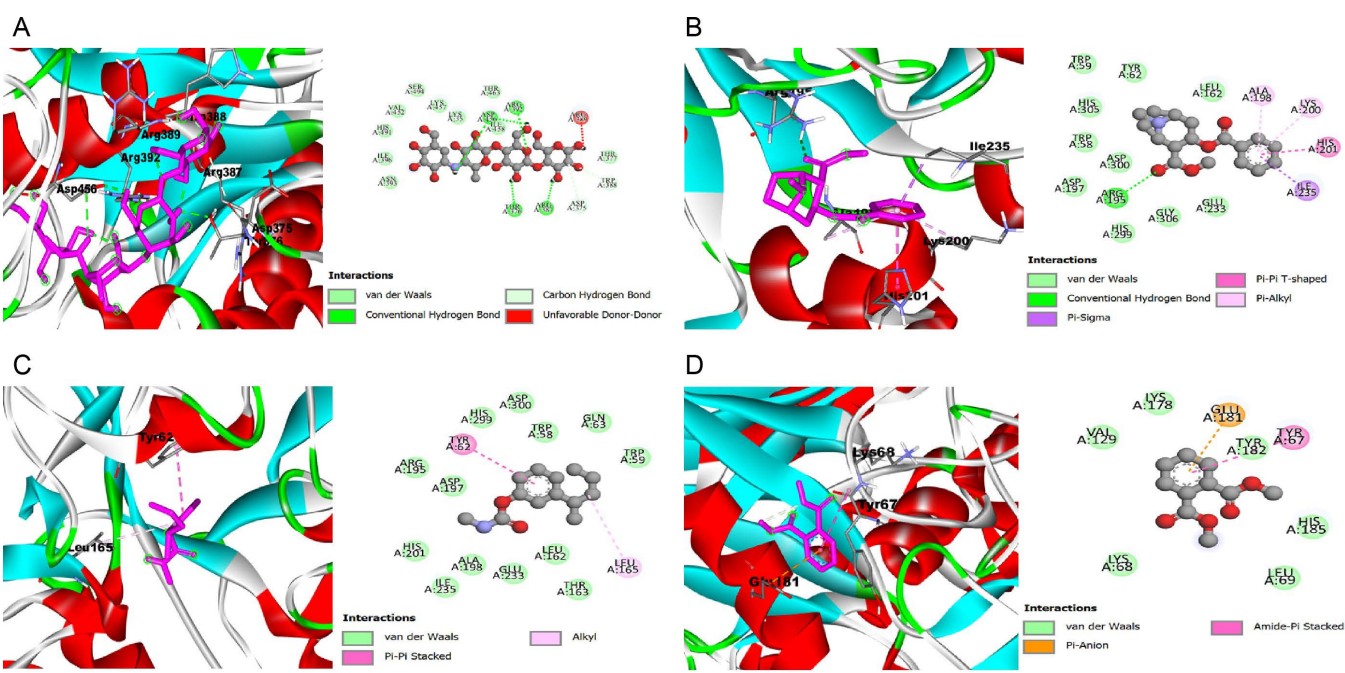

**Fig 3. The three and two-dimensional (3D and 2D) view of the molecular interactions of amino acid residues of α-amylase with Acarbose(A),** Pseudo-cocaine**(B) M-(1-Methylbutyl)phenyl methylcarbamate (C) Dimethyl phthalate (D).**

The significance of anti-inflammatory drugs in the treatment of diabetes should not be overlooked [44]. The study found that extracts from T. cattapa, P. americana, and *T. conophorum* had anti-inflammatory activity that varied according on concentration. Notably, the extract from *T. cattapa* shown a greater suppression of RBC hemolysis, suggesting stronger anti-inflammatory capabilities in these plant species. Our results are consistent with the findings of Iheagwam et al[18]

Minerals are crucial for the proper functioning of the body since they play a vital role in several biochemical events. They operate as stabilising components of enzymes and proteins and also serve as cofactors for numerous enzymes[45]. Copper enhances the body's general metabolic processes and promotes gastrointestinal well-being, particularly advantageous for individuals with diabetes. Copper aids in the reduction of blood glucose spikes at specific periods. Magnesium is an essential component for facilitating the transport of glucose into cells and for the metabolism of carbohydrates. It plays a role in the cellular processes related to insulin. Inadequate consumption of magnesium increases the likelihood of developing diabetes. A lack of magnesium hampers the ability of cells to protect themselves against damage produced by oxidation. This, in turn, reduces the body's ability to withstand the oxidative stress induced by diabetes, leading to a faster development of diseases associated with diabetes. Calcium homeostasis significantly influences both insulin resistance and secretion. Diabetes disrupts calcium homeostasis, leading to dysfunctional cell control in erythrocytes, cardiac muscles, platelets, and skeletal muscles. The disrupted homeostasis is worrisome as it may play a substantial role in the control of appropriate insulin secretion and function [46]. Manganese deficiency is prevalent among individuals with diabetes, and there is speculation in certain quarters that it may contribute to the development of diabetes. Manganese has a crucial role as a cofactor in the enzymatic regulation of glucose metabolism. The use of insulin in the treatment regimen used by many individuals with diabetes might lead to potassium shortage.

**Table 14. Binding affinities (ΔG in kcal/mol) of Alpha-glucosidase with bioactive components of *P. americana*, *T. conophorum* and *T. cattapa*.**

| | Compounds name | Source | CID number | Alpha -glucosidase (5NN5) |
|---|---|---|---|---|
| 1. | 1-deoxynojirimycin | Co-crystalline compound | 29435 | −5.3 |
| 2. | Dimethyl phthalate | *T. cattapa* | 8554 | −5.3 |
| 3. | O-xylene | *T. conophorum* | 7237 | −5.4 |
| 4. | Decan 2,2,4 trimethyl | *T. cattapa* | 545792 | −5 |
| 5. | 9,12,15-octadecatrienal | *T. cattapa* | 5283384 | −4.5 |
| 6. | 2-chloro-3-oxohexanedioic acid | *P. americana,* | 443963 | −5.3 |
| 7. | Acarbose | Standard | 41774 | −7.8 |
| 8. | Pseudococaine | *T. cattapa* | 2826 | −6.6 |
| 9. | Methyl 2-chloro-5-nitrobenzoate | *P. americana,* | 22754 | −5.4 |
| 10. | M-(1-methylbutyl)phenyl methylcarbamate | *T. cattapa* | 16787 | −6.5 |
| 11. | Cyclohexane | *T. cattapa* | 12762849 | −5 |
| 12. | 1,4-dimethylcyclohexane | *T. conophorum* | 11523 | −5.2 |

**Table 15. Binding affinities (ΔG in kcal/mol) of Alpha-amylase with bioactive components of *P. americana, T. conophorum* and *T. cattapa*.**

| | Compounds name | Plant source | CID number | Alpha-amylase (3BAI) |
|---|---|---|---|---|
| 1. | Nitrate | Co-crystalline compound | 943 | −3.4 |
| 2. | Dimethyl phthalate | *T. cattapa* | 8554 | −6 |
| 3. | O-xylene | *T. conophorum* | 7237 | −4.9 |
| 4. | Decan 2,2,4 trimethyl | *T. cattapa* | 545792 | −4.8 |
| 5. | 9,12,15-octadecatrienal | *T. cattapa* | 5283384 | −5 |
| 6. | 2-chloro-3-oxohexanedioic acid | *P. americana,* | 443963 | −5.4 |
| 7. | Acarbose | Standard | 41774 | −8.4 |
| 8. | Pseudococaine | *T. cattapa* | 2826 | −7 |
| 9. | Methyl 2-chloro-5-nitrobenzoate | *P. americana,* | 22754 | −5.8 |
| 10. | M-(1-Methylbutyl)phenyl methylcarbamate | *T. cattapa* | 16787 | −6.2 |
| 11. | Cyclohexane | *T. cattapa* | 12762849 | −4.9 |
| 12. | 1,4-dimethylcyclohexane | *T. conophorum* | 11523 | −4.7 |

Adding potassium to a nutritious diet can enhance insulin sensitivity and the efficacy of the hormone for those with diabetes. According to our findings, the seed of *Persea americana* exhibited the greatest levels of minerals, including Copper, Manganese, Magnesium, Potassium, and Calcium. These results align with the findings reported by Chakradhari et al. [47]

The study examined the proximate composition of three plant samples, including their moisture content, crude fibres, carbohydrates, lipids, and ash. The *T. cattapa* kernel exhibited larger percentages of crude fibres, carbs, protein, lipids, and ash compared to the *P. americana* seed and *T. conophorum* nut. This suggests that the *T. cattapa* kernel has the potential to be used as a remedy for managing diabetes. Fibres are components of fruits, cereals, and vegetables that can be either digested or absorbed by the human body. Dietary fibres primarily serve to decelerate the pace at which glucose is absorbed into the circulation, thus diminishing the

likelihood of hyperglycemia [48, 49]. The ash level of a plant-based diet is determined by the mineral components it contains. Dietary ash has been found to be beneficial in regulating and preserving the acid-base equilibrium of the bloodstream, as well as managing the state of hyperglycemia.. Vitamins are vital organic compounds that an organism need in tiny amounts for optimal metabolic processes, either as coenzymes or precursors [50]. They have a crucial role in the treatment of diabetes due to their well-established anti-inflammatory functions [50]. They can enhance the general functioning of the immune system and inhibit the progression of diabetes [50]. Water-soluble vitamins have the ability to dissolve in water, but they lack the capacity to be readily stored inside the body. These include vitamin B1, or thiamine, which helps enhance glucose metabolism, lower the risk of cardiovascular complications associated with diabetes, and alleviate stress. Riboflavin, generally referred to as Vitamin B2, might mitigate the likelihood of developing type 2 diabetes in those with pre-diabetes. Additionally, it can alleviate any inflammation resulting from oxidative stress. It facilitates the conversion of energy derived from carbohydrates into a usable form of fuel. Vitamin B3, often known as Niacin, aids in the regulation of cholesterol levels and mitigates cardiovascular issues associated with diabetes. Pantothenic acid, commonly known as Vitamin B5, is crucial for the production of fatty acids and plays a vital role in enhancing overall metabolism. Additionally, it aids the body in converting energy. Vitamin B6, also known as Pyridoxine, facilitates the release of stored carbohydrates in the body and converts them into energy. It enhances nerve health, thereby reducing the risk of diabetic neuropathy and diabetic retinopathy. Consistent consumption of this vitamin can also slow down the advancement of diabetic nephropathy. Biotin, commonly known as Vitamin B7, is essential for the metabolism of glucose, amino acids, and fatty acids. Additionally, it enhances the body's utilisation of glucose. It enhances glucose storage and mitigates the likelihood of insulin resistance. Folate, often known as Vitamin B9, decreases the likelihood of cardiovascular illnesses in those diagnosed with type 2 diabetes. Vitamin B12 (Cobalamin) mitigates the risk or averts the beginning of diabetic neuropathy. Additionally, it aids in alleviating discomfort associated with health disorders connected to diabetes. Vitamin C, also known as ascorbic acid, is a potent antioxidant that enhances the body's immune system. It reduces the accumulation of sorbitol sugar in nerve, eye, and kidney cells, which can lead to damage. This helps prevent the development of diabetic neuropathy and diabetic retinopathy. Additionally, it aids in lowering blood pressure and mitigates the likelihood of cardiovascular health issues associated with diabetes. Fat-soluble vitamins are hydrophobic vitamins that are insoluble in water. Vitamin A is essential for maintaining good eyesight and preventing visual problems caused by diabetes. It also supports the secretion of insulin by beta cells, enhances metabolic activities, and regulates the appropriate release of insulin. Vitamin D enhances the synthesis of peptides that eradicate pathogens, bacteria, and viruses, hence decreasing the likelihood of developing gum disease associated with diabetes or diabetic ulcers. Vitamin D enhances overall bodily function and fortifies gums and teeth, hence mitigating oral health concerns associated with diabetes. Vitamin E functions as an antioxidant, protecting cells from harm. It enhances the body's ability to regulate glucose levels and reduces the likelihood of free radical-induced damage to blood vessels and nerves, which is commonly associated with diabetes. Additionally, it has the potential to reverse nerve damage caused by diabetes and lowers the risk of diabetic cataracts and atherosclerosis. Vitamin K has a crucial role in the activation of proteins and calcium, which in turn can lower the likelihood of experiencing fractures [32]. The *T. conophorum* nut exhibited the greatest levels of fat-soluble and water-soluble vitamins, which aligns with the findings of Asouzu et al.[32].

The gas chromatography-mass spectroscopy study detected the existence of bioactive chemicals in the three plant samples. The *T. conophorum* nut contained a total of sixty-five compounds, with the greatest concentration found in three compounds: 2-Hexenal, 1,3-hexadien-5-yne, and

Cycloheptane. 2-hexenal is a volatile chemical generated by green plants. It is a tiny molecule that plays a crucial function in controlling plant development and enhancing resilience to different environmental stressors. It clearly hinders the growth of plant roots and the ability to defend against bacterial infection and herbivorous eating. 2-hexenal also functions as a "messenger" in conveying defence signals between plants [51]. Cycloheptane is a kind of cycloalkane that serves as a nonpolar solvent in the chemical industry and plays a role as an intermediary in the production of various compounds and medicinal medications. The substance can be bothersome to the eyes and may lead to respiratory depression if breathed in significant amounts [52]. The *T. cattapa* kernel contained a total of forty-six compounds, with the highest concentration found in three compounds: 1-ethyl-3-metyl, 1,3-dimetyl, and 1-methylbutyl. On the other hand, the *P. americana* seed had thirteen compounds, with the highest concentration found in three compounds: 2–2-chloro ethoxy-ethoxyl phenol. Propanoic acid is naturally present in certain food products, such as milk, cheese, and yoghurt, as a result of bacterial fermentation, primarily by propionibacteria. It is also used as a preservative (E280) in food items due to its antifungal and antibacterial properties [52]. Methylene chloride is an organic compound containing chlorine. This transparent and highly volatile liquid, which has a chloroform-like and pleasant scent, is commonly employed as a solvent. While it does not mix with water, it has a minor polarity and can mix with many organic solvents. This substance is utilised for the purpose of removing caffeine from coffee and tea, as well as for creating extracts of hops and other flavourings [53].

The molecular docking investigation of the bioactive chemicals from three plant samples, namely *T. conophorum* nut, *P. americana* seed, and *T. cattapa* kernel, revealed that these plants exhibit different binding affinities towards both alpha-glucosidase and amylase. Upon comparing their scores, it was seen that acarbose exhibited more affinities than the other. The binding activity of bioactive chemicals from *T. cattapa* kernel was observed to be greater than that of *P. americana* seed for alpha-amylase. A little correlation was noted between this finding and the in vitro inhibitory activity conducted in this investigation. The alpha-glucosidase enzyme had a comparable outcome, wherein the *T. cattapa* kernel displayed a greater affinity for binding, while both the *T. conophorum* nut and *P. americana* seed exhibited equal binding affinities. Upon comparison, it was shown that the bioactive chemicals exhibited a greater affinity as compared to their co-crystalline inhibitors. Certain chemicals in these plants engaged with the amino acids in the binding pockets, exhibiting significant resemblances to the conventional medication (acarbose).

## Conclusion

This study examined the in vitro antidiabetic, antioxidant, and anti-inflammatory activities of plant samples from *T. conophorum* nut, *P. americana* seed, and *T. cattapa* kernel. Additionally, in-silico analysis was conducted. The in-vitro study findings indicated that the *P. americana* seed had the most effective antidiabetic qualities among the three plants. Additionally, the in-silico analysis demonstrated that the *T. cattapa* kernel had a greater binding affinity towards the diabetic enzymes. The kernel of *T. cattapa* demonstrated superior antioxidant and anti-inflammatory activities, mineral content, and proximate proportion. The *T. conophorum* nut is abundant in both water-soluble and fat-soluble vitamins. These plant components have pharmacological and nutritional qualities that can be utilised to create medications for manageent of diabetes and its associated complications.

## Supporting information

**S1 Raw Data. Raw data of the experiments performed.**
(ZIP)

## Author contributions

**Conceptualization:** Iwara Aripko Iwara, Daniel Ejim Uti, Item Justin Atangwho.

**Data curation:** Efah Denis Eyong, Eyuwa Ignatius Agwupuye, Abdulhakeem Rotimi Agboola, Wilson Arong Obio.

**Formal analysis:** Iwara Aripko Iwara, Wilson Arong Obio, Esther Ugo Alum.

**Investigation:** Efah Denis Eyong, Eyuwa Ignatius Agwupuye, Abdulhakeem Rotimi Agboola.

**Supervision:** Iwara Aripko Iwara, Item Justin Atangwho.

**Validation:** Abdulhakeem Rotimi Agboola, Daniel Ejim Uti, Wilson Arong Obio, Esther Ugo Alum, Item Justin Atangwho.

**Visualization:** Eyuwa Ignatius Agwupuye.

**Writing – original draft:** Efah Denis Eyong, Eyuwa Ignatius Agwupuye, Abdulhakeem Rotimi Agboola, Wilson Arong Obio.

**Writing – review & editing:** Iwara Aripko Iwara, Daniel Ejim Uti, Esther Ugo Alum, Item Justin Atangwho.

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
