## [Decision Letter · Decision Letter 0]

22 Oct 2024

PONE-D-24-37433In vitro and in silico pharmaco-nutritional assessments of some lesser-known Nigerian nuts: Persea americana, Tetracarpidium conophorum, and Terminalia catappaPLOS ONE

Dear Dr. Uti,

Thank you for submitting your manuscript to PLOS ONE. After careful consideration, we feel that it has merit but does not fully meet PLOS ONE’s publication criteria as it currently stands. Therefore, we invite you to submit a revised version of the manuscript that addresses the points raised during the review process.

We look forward to receiving your revised manuscript.

Kind regards,

Oluwafemi Adeleke Ojo, Ph.D

Academic Editor

PLOS ONE

Journal Requirements:

2. We suggest you thoroughly copyedit your manuscript for language usage, spelling, and grammar. If you do not know anyone who can help you do this, you may wish to consider employing a professional scientific editing service. The American Journal Experts (AJE) (https://www.aje.com/) is one such service that has extensive experience helping authors meet PLOS guidelines and can provide language editing, translation, manuscript formatting, and figure formatting to ensure your manuscript meets our submission guidelines. Please note that having the manuscript copyedited by AJE or any other editing services does not guarantee selection for peer review or acceptance for publication. Upon resubmission, please provide the following: ● The name of the colleague or the details of the professional service that edited your manuscript ● A copy of your manuscript showing your changes by either highlighting them or using track changes (uploaded as a *supporting information* file) ● A clean copy of the edited manuscript (uploaded as the new *manuscript* file)

3. In the online submission form, you indicated that “Data is available from the corresponding author on request”.

All PLOS journals now require all data underlying the findings described in their manuscript to be freely available to other researchers, either 1. In a public repository, 2. Within the manuscript itself, or 3. Uploaded as supplementary information. This policy applies to all data except where public deposition would breach compliance with the protocol approved by your research ethics board. If your data cannot be made publicly available for ethical or legal reasons (e.g., public availability would compromise patient privacy), please explain your reasons on resubmission and your exemption request will be escalated for approval.

Additional Editor Comments:

All comments raised by the reviewers is important and must be addressed before this manuscript can be considered for publication.

Reviewers' comments:

Reviewer's Responses to Questions

**Comments to the Author**

1. Is the manuscript technically sound, and do the data support the conclusions?

Reviewer #1: Partly

2. Has the statistical analysis been performed appropriately and rigorously? 

Reviewer #1: Yes

3. Have the authors made all data underlying the findings in their manuscript fully available?

Reviewer #1: Yes

4. Is the manuscript presented in an intelligible fashion and written in standard English?

Reviewer #1: Yes

5. Review Comments to the Author

Reviewer #1: Title: In vitro and in silico pharmaco-nutritional assessments of some lesser-known Nigerian nuts: Persea americana, Tetracarpidium conophorum, and Terminalia catappa

Manuscript Number: PONE-D-24-37433

The study aims to evaluate the antidiabetic, antioxidant, and anti-inflammatory effects of extracts from Nigerian nuts using in vitro and in silico approaches. While the research is potentially valuable in identifying new bioactive compounds, there are several critical areas that need improvement in terms of clarity, methodological rigor, and scientific reporting.

Strengths:

1.Relevance: The topic is relevant as it explores the potential medicinal benefits of underutilized nuts from Nigeria, contributing to the broader field of plant-based therapeutic research.

2.Multi-disciplinary Approach: The combination of in vitro and in silico assessments strengthens the study by offering insights from different perspectives.

3.Diversity of Assays: The study employs a variety of assays (antioxidant, anti-inflammatory, α-glucosidase inhibition, and α-amylase inhibition) to comprehensively assess the medicinal properties of the nuts.

Areas of Improvement:

1.Abstract and Introduction:

oThe abstract is packed with results but lacks a clear problem statement and objectives. The introduction should better frame the gap in the literature and the rationale for selecting these particular nuts.

oThe inclusion of basic nutritional qualities in the abstract feels secondary and could distract from the more critical pharmacological outcomes.

2.Methodology:

oSome protocols lack sufficient detail. For example, in the α-amylase inhibitory activity section, the use of GOD reagent is not adequately explained in the context of this assay, and there is no clarity on whether controls were sufficiently employed.

oThe methods for mineral analysis and vitamin analysis are not thoroughly described. It is unclear whether standardized reference materials were used, making it difficult to assess the accuracy of the mineral and vitamin content reported.

oThe molecular docking method lacks sufficient detail on the choice of ligands. The rationale for selecting specific bioactive compounds for docking studies should be better justified based on previous literature or preliminary screening results.

3.Data Presentation and Results:

oThe results section is overly descriptive and lacks interpretation. It is difficult to understand the significance of certain results, such as why particular compounds are highlighted in molecular docking or what the practical implications of the in vitro assays are.

oThe tables and figures are somewhat cluttered, and it is challenging to draw clear comparisons between the three nuts. Simplifying the presentation of results or creating better visual aids would improve clarity.

4.Discussion:

oThe discussion lacks depth in connecting the experimental outcomes with broader implications for human health. While the study shows that the extracts possess bioactivity, the potential therapeutic applications and limitations are not fully explored.

oThere is limited mention of the limitations of the in vitro models used and no discussion of how these results translate to in vivo efficacy or human relevance. Additionally, there is no exploration of potential side effects or toxicity.

oThe relevance of the docking results is not well integrated into the broader findings. The discussion should elaborate on whether the docking results suggest a feasible mechanism of action.

5.Scientific Rigor:

oThe statistical methods used are appropriate, but more clarity is needed on how biological replicates were handled across different assays.

oThe study could benefit from validating the in vitro results with additional models (e.g., in vivo or ex vivo models) to strengthen the conclusions.

oThere are a few instances where the manuscript's language lacks precision (e.g., ambiguous terms such as “substantial inhibition”). Precision in wording would improve the paper’s readability and reliability.

6.Literature Review:

oThe manuscript does not cite enough recent studies, particularly in molecular docking or in the evaluation of antidiabetic effects in plant extracts. Incorporating more up-to-date references would strengthen the background and discussion sections.

7.Ethics and Data Sharing:

oThe ethics statement is missing, though it is noted that human blood cells were used in the RBC assays. Approval from an appropriate ethics board should be mentioned if applicable.

Conclusion:

While the study provides initial evidence that these nuts possess useful pharmacological properties, it is far from complete. The results are promising but lack proper contextualization and interpretation. More rigor in both the presentation of methods and the discussion of findings is required.

6. PLOS authors have the option to publish the peer review history of their article (what does this mean? ). If published, this will include your full peer review and any attached files.

**Do you want your identity to be public for this peer review?** For information about this choice, including consent withdrawal, please see our Privacy Policy .

Reviewer #1: **Yes: ** Adewale Oluwaseun Fadaka

---

## [Author Response · Author response to Decision Letter 1]

10 Dec 2024

Dear Editor in Chief,

Plos One,

Submission of Revised manuscript

We appreciate you and the reviewers for your precious time in reviewing our paper and providing valuable comments. The authors have carefully considered the comments and tried our best to address every one of them. Below we provide the point-by-point responses. All modifications in the manuscript have been highlighted in yellow colour

Title: In vitro and in silico pharmaco-nutritional assessments of some lesser-known Nigerian nuts: Persea americana, Tetracarpidium conophorum, and Terminalia catappa

Manuscript Number: PONE-D-24-37433

Reviewer #1: Title: In vitro and in silico pharmaco-nutritional assessments of some lesser-known Nigerian nuts: Persea americana, Tetracarpidium conophorum, and Terminalia catappa

Manuscript Number: PONE-D-24-37433

The study aims to evaluate the antidiabetic, antioxidant, and anti-inflammatory effects of extracts from Nigerian nuts using in vitro and in silico approaches. While the research is potentially valuable in identifying new bioactive compounds, there are several critical areas that need improvement in terms of clarity, methodological rigor, and scientific reporting.

Strengths:

1.Relevance: The topic is relevant as it explores the potential medicinal benefits of underutilized nuts from Nigeria, contributing to the broader field of plant-based therapeutic research.

Author’s response: Thank you very much for your thoughtful feedback and for recognizing the relevance of our study. We are glad that the potential medicinal benefits of these underutilized Nigerian nuts resonate with you. We hope that our findings contribute meaningfully to advancing plant-based therapeutic research and inspire further exploration in this field. Your positive comments are greatly appreciated and motivate us to continue exploring the valuable resources within underutilized plants.

2.Multi-disciplinary Approach: The combination of in vitro and in silico assessments strengthens the study by offering insights from different perspectives.

Author’s response: Thank you for your positive feedback on the multi-disciplinary approach in our study. We appreciate your recognition of the value of combining in vitro and in silico assessments, as we aimed to provide a more comprehensive understanding by integrating insights from different perspectives. This approach was intended to enhance the robustness of our findings, and we are glad that this aspect resonated with you. Your encouraging comment reinforces our commitment to employing multi-faceted methodologies in future research

3.Diversity of Assays: The study employs a variety of assays (antioxidant, anti-inflammatory, α-glucosidase inhibition, and α-amylase inhibition) to comprehensively assess the medicinal properties of the nuts.

Author’s response: Thank you very much for your positive feedback! We are glad that you appreciate the diversity of assays included in our study. Our aim was to provide a comprehensive evaluation of the medicinal properties of the nuts, and we believe that employing multiple assays, such as antioxidant, anti-inflammatory, α-glucosidase, and α-amylase inhibition, allowed us to capture a holistic view of their potential health benefits. Your recognition of this approach is greatly encouraging and motivates us to continue exploring diverse methods to assess bioactive compounds more effectively.

Areas of Improvement:

1.Abstract and Introduction:

oThe abstract is packed with results but lacks a clear problem statement and objectives. The introduction should better frame the gap in the literature and the rationale for selecting these particular nuts.

Author’s response: The revised abstract has integrated a clearer problem statement, rationale for selecting these nuts, and a more structured presentation of the study objectives and findings.

oThe inclusion of basic nutritional qualities in the abstract feels secondary and could distract from the more critical pharmacological outcomes.

Author’s response: We have revised the abstract to prioritize the pharmacological outcomes, as suggested, and have minimized the mention of basic nutritional qualities to maintain focus on the primary objectives of the study.

2.Methodology:

oSome protocols lack sufficient detail. For example, in the α-amylase inhibitory activity section, the use of GOD reagent is not adequately explained in the context of this assay, and there is no clarity on whether controls were sufficiently employed.

Author’s response: We have addressed your concerns by adding more details to the protocol in the revised manuscript. Specifically, we have clarified the role of the GOD reagent in the α-amylase inhibitory activity assay, as well as its relevance to the assay’s mechanism. Additionally, we have provided further explanation on the controls employed in the study to ensure accuracy and reproducibility of the results.

oThe methods for mineral analysis and vitamin analysis are not thoroughly described. It is unclear whether standardized reference materials were used, making it difficult to assess the accuracy of the mineral and vitamin content reported.

Author’s response: Thank you for your insightful feedback. Standard methods were indeed used for both mineral and vitamin analyses, and we intentionally summarized the description of these methods to streamline the manuscript and avoid unnecessary bulk. To maintain clarity and allow for reproducibility, we referenced the primary sources detailing these standard procedures, however, based on your request, we have given a more detailed explanations to the process, and avoiding unnecessary bulkiness.

oThe molecular docking method lacks sufficient detail on the choice of ligands. The rationale for selecting specific bioactive compounds for docking studies should be better justified based on previous literature or preliminary screening results.

Author’s response: Thank you for your valuable feedback. The manuscript has been revised to include comprehensive details on the molecular docking methodology. Specifically, we have elaborated on the rationale for selecting certain bioactive ligands, including insights from previous literature and results that guided our choices. Additionally, we have provided detailed information on grid box selection to clarify our docking parameters.

3. Data Presentation and Results:

oThe results section is overly descriptive and lacks interpretation. It is difficult to understand the significance of certain results, such as why particular compounds are highlighted in molecular docking or what the practical implications of the in vitro assays are.

Author’s response: We have added more context to highlight the significance of the selected compounds in the molecular docking studies, including their potential interactions and relevance to our research objectives. Additionally, we have elaborated on the practical implications of the in vitro assays, providing clearer connections to how these results may impact future therapeutic applications.

oThe tables and figures are somewhat cluttered, and it is challenging to draw clear comparisons between the three nuts. Simplifying the presentation of results or creating better visual aids would improve clarity.

Author’s response: We have simplified the tables and figures to enhance clarity and ensure more straightforward comparisons between the three nuts. The updated visuals now provide a cleaner, more organized presentation, making it easier to interpret the results.

4. Discussion:

oThe discussion lacks depth in connecting the experimental outcomes with broader implications for human health. While the study shows that the extracts possess bioactivity, the potential therapeutic applications and limitations are not fully explored.

Author’s response: Thank you for your feedback. In response, we have expanded the Discussion section to provide a more in-depth analysis, connecting our experimental outcomes with broader implications for human health. We have elaborated on the potential therapeutic applications of the bioactive extracts, discussing how they might be applied in clinical settings while also addressing the limitations of our study. oThere is limited mention of the limitations of the in vitro models used and no discussion of how these results translate to in vivo efficacy or human relevance. Additionally, there is no exploration of potential side effects or toxicity.

Author’s response:

oThe relevance of the docking results is not well integrated into the broader findings. The discussion should elaborate on whether the docking results suggest a feasible mechanism of action.

Author’s response: We appreciate your insights on improving the integration of the docking results into the broader findings. In the revised manuscript, we have addressed this concern in lines 143 to 157. We have enhanced the discussion to better connect the docking results with the overall study findings, elaborating on how these results suggest a feasible mechanism of action. Specifically, we discuss how the docking interactions correlate with key binding sites and pathways relevant to the target compound's biological activity, thereby supporting a potential mechanism of action.

5.Scientific Rigor:

oThe statistical methods used are appropriate, but more clarity is needed on how biological replicates were handled across different assays.

Author’s response: Thank you for acknowledging our efforts, we have made clarifications on handing of replicates

oThe study could benefit from validating the in vitro results with additional models (e.g., in vivo or ex vivo models) to strengthen the conclusions.

Author’s response: This will be done in the next phase of the study, where we intend to further investigations on this baseline study

oThere are a few instances where the manuscript's language lacks precision (e.g., ambiguous terms such as “substantial inhibition”). Precision in wording would improve the paper’s readability and reliability.

Author’s response: We have proofread and edit for grammatical accuracy, and remove ambiguous terms

6.Literature Review:

oThe manuscript does not cite enough recent studies, particularly in molecular docking or in the evaluation of antidiabetic effects in plant extracts. Incorporating more up-to-date references would strengthen the background and discussion sections.

Author’s response: we have incorporated additional recent studies, particularly in the areas of molecular docking and the evaluation of antidiabetic effects in plant extracts. These updates enhance the background and discussion sections, providing a more comprehensive overview of current research and supporting the study’s relevance within the field. We believe these additions address your concern and strengthen the manuscript's foundation

7.Ethics and Data Sharing:

oThe ethics statement is missing, though it is noted that human blood cells were used in the RBC assays. Approval from an appropriate ethics board should be mentioned if applicable.

Author’s response: The blood sample was obtained from rats, and not human. Moreover, The animal handling protocol was in line with the guidelines of the National Institute of Health (NIH) publication (1985) for laboratory animal care and use. Moreover, ethical approval for the use of animals in this study was obtained from the Faculty of Basic Medical Sciences Animal Research Ethics Committee (FAREC- FBMS, Approval number: 032BCH3319), University of Calabar, Nigeria

Conclusion:

While the study provides initial evidence that these nuts possess useful pharmacological properties, it is far from complete. The results are promising but lack proper contextualization and interpretation. More rigor in both the presentation of methods and the discussion of findings is required.

Author’s response: We have carefully revised the manuscript to address the points you raised. Specifically, we have strengthened the contextualization and interpretation of our results, providing a more rigorous discussion of the pharmacological properties of the nuts in question. Additionally, we have refined our presentation of the methods to ensure clarity and thoroughness

---

## [Decision Letter · Decision Letter 1]

7 Feb 2025

In vitro and in silico pharmaco-nutritional assessments of some lesser-known Nigerian nuts: Persea americana, Tetracarpidium conophorum, and Terminalia catappa

PONE-D-24-37433R1

Dear Dr. Uti,

We’re pleased to inform you that your manuscript has been judged scientifically suitable for publication and will be formally accepted for publication once it meets all outstanding technical requirements.

Kind regards,

Oluwafemi Adeleke Ojo, Ph.D

Academic Editor

PLOS ONE

Additional Editor Comments (optional):

Reviewers' comments:

Reviewer's Responses to Questions

**Comments to the Author**

1. If the authors have adequately addressed your comments raised in a previous round of review and you feel that this manuscript is now acceptable for publication, you may indicate that here to bypass the “Comments to the Author” section, enter your conflict of interest statement in the “Confidential to Editor” section, and submit your "Accept" recommendation.

Reviewer #2: All comments have been addressed

Reviewer #3: All comments have been addressed

2. Is the manuscript technically sound, and do the data support the conclusions?

Reviewer #2: Yes

Reviewer #3: Yes

3. Has the statistical analysis been performed appropriately and rigorously? 

Reviewer #2: Yes

Reviewer #3: Yes

4. Have the authors made all data underlying the findings in their manuscript fully available?

Reviewer #2: Yes

Reviewer #3: Yes

5. Is the manuscript presented in an intelligible fashion and written in standard English?

Reviewer #2: Yes

Reviewer #3: Yes

6. Review Comments to the Author

Reviewer #2: (No Response)

Reviewer #3: This research article seeks to breach the gap between invitro and insilico studies, which will increase the potential of using these nuts for medicinal purposes. Although it should be noted that no dual publication of this article is accepted.

7. PLOS authors have the option to publish the peer review history of their article (what does this mean? ). If published, this will include your full peer review and any attached files.

**Do you want your identity to be public for this peer review?** For information about this choice, including consent withdrawal, please see our Privacy Policy .

Reviewer #2: **Yes: ** Odunayo Anthonia TAIWO

Reviewer #3: No

---

## [Editor Report · Acceptance letter]

PONE-D-24-37433R1

PLOS ONE

Dear Dr. Uti,

I'm pleased to inform you that your manuscript has been deemed suitable for publication in PLOS ONE. Congratulations! Your manuscript is now being handed over to our production team.

Kind regards,

on behalf of

Dr. Oluwafemi Adeleke Ojo

Academic Editor

PLOS ONE